# Molecular and Clinical Characteristics of Different Toxicity Rates in Anti-CD19 Chimeric Antigen Receptor T Cells: Real-World Experience

**DOI:** 10.3390/cancers15174253

**Published:** 2023-08-25

**Authors:** E. Gavriilaki, D. Mallouri, Z. Bousiou, C. Demosthenous, A. Vardi, P. Dolgyras, I. Batsis, E. Stroggyli, P. Karvouni, M. Masmanidou, M. Gavriilaki, A. Bouinta, S. Bitsianis, N. Kapravelos, M. Bitzani, G. Vasileiadou, E. Yannaki, D. Sotiropoulos, S. Papagiannopoulos, D. Kazis, V. Kimiskidis, A. Anagnostopoulos, I. Sakellari

**Affiliations:** 1Hematology Department and Bone Marrow Transplant (BMT) Unit, G. Papanicolaou Hospital, 57010 Thessaloniki, Greece; dmallouri@gmail.com (D.M.); boussiou_z@hotmail.com (Z.B.); christosde@msn.com (C.D.); anna_vardi@yahoo.com (A.V.); panadolg@gmail.com (P.D.); iobats@yahoo.gr (I.B.); evlabia1@gmail.com (E.S.); mariannareti@gmail.com (M.M.); alexis.menia@gmail.com (A.B.); eyannaki@uw.edu (E.Y.); dsotiro@otenet.gr (D.S.); achanagh@gmail.com (A.A.); ioannamarilena@gmail.com (I.S.); 2Medical School, Aristotle University of Thessaloniki, 54636 Thessaloniki, Greece; paraskeu98@gmail.com; 31st Department of Neurology, AHEPA University Hospital, Aristotle University of Thessaloniki, 54636 Thessaloniki, Greece; mariagavri6@yahoo.gr (M.G.); kimiskid@auth.gr (V.K.); 4Department of Surgery, G. Papanicolaou Hospital, Aristotle University of Thessaloniki, 54636 Thessaloniki, Greece; sbitsiani@gmail.com; 51st Intensive Care Unit, G. Papanicolaou Hospital, 57010 Thessaloniki, Greece; kapravelos@gmail.com (N.K.); malamatimsf@yahoo.gr (G.V.); 62nd Intensive Care Unit, G. Papanicolaou Hospital, 57010 Thessaloniki, Greece; bitmilly@gmail.com (M.B.); ubic1@otenet.gr (S.P.); 73rd Department of Neurology, AHEPA University Hospital, Aristotle University of Thessaloniki, 54636 Thessaloniki, Greece; dimitrios.kazis@gmail.com

**Keywords:** CAR T cells, toxicity, brexucabtagene autoleucel, tisagenlecleucel, axicabtagene ciloleucel, cytokine release syndrome (CRS), immune-effector-cell-associated neurotoxicity syndrome (ICANS)

## Abstract

**Simple Summary:**

Our real-world experience confirmed that commercial CART therapy can be administered with minimal toxicity. The early referral of patients with low tumor burdens is important as CAR T indications continue to expand. Furthermore, close monitoring and the early recognition of side effects are beneficial to preventing major toxicities and may potentially expand the use of CAR T therapy.

**Abstract:**

Commercially available anti-CD19 chimeric antigen receptor T cells (CARΤ cells) have offered long-term survival to a constantly expanding patient population. Given that novel toxicities including cytokine release syndrome (CRS) and neurotoxicity (ICANS) have been observed, we aimed to document the safety and toxicity of this treatment in a real-world study. We enrolled 31 adult patients referred to our center for CAR T therapy. Tisagenlecleucel was infused in 12 patients, axicabtagene ciloleucel in 14, and brexucabtagene autoleucel in 5. Cytokine release syndrome was noted in 26 patients while neurotoxicity was observed in 7. Tocilizumab was administered for CRS in 18 patients, along with short-term, low-dose steroid administration in one patient who developed grade III CRS and, subsequently, grade I ICANS. High-dose steroids, along with anakinra and siltuximab, were administered in only two MCL patients. With a median follow-up time of 13.4 months, nine patients were then in CR. The progression-free (PFS) and overall survival (OS) rates were 41.2% and 88.1% at one year, respectively. MCL diagnosis, which coincides with the administration of brexucabtagene autoleucel, was the only factor to be independently associated with poor OS (*p* < 0.001); meanwhile, increased LDH independently predicted PFS (*p* = 0.027).In addition, CRP at day 14 was associated with a poor OS (*p* = 0.001). Therefore, our real-world experience confirmed that commercial CAR T therapy can be administered with minimal toxicity.

## 1. Introduction

Novel chimeric antigen receptor (CAR) T cell therapies have been initially approved for use in treating acute lymphoblastic leukemia (ALL) and relapsed/refractory aggressive B non-Hodgkin lymphoma (NHL). CAR T cells are manufactured via the isolation and genetic engineering of autologous T cells in order to express a certain modified T cell receptor) [1,2,3].First-generation CAR T cells consist of a single-chain variable fragment antigen-recognition domain, a transmembrane domain, and a T-cell activation domain (CD3-derived). Second generation cells have an additional co-stimulatory domain (CD28 or 4-1BB), while third generation cells have two co-stimulatory domains (for example, both CD28 and 4-1BB) [3]. Two commercial biosynthetic CD19 second-generation CART cell products have been in use in recent years: tisagenlecleucel and axicabtagene ciloleucel [4]. Tisagenlecleucel is used to treat relapsed/refractory B-ALL patients up to 25 years old [5] and relapsed or refractory diffuse large B-cell lymphoma (DLBCL) patients [6,7]. On the other hand, axicabtagene ciloleucel is only used for the treatment of refractory large B-cell lymphoma [8,9].Brexucabtagene autoleucel (Tecartus) is anothersecond-generationCD19 CAR T cell product that was recently approved for treating relapsed or refractory mantle cell lymphoma (MCL) [10,11,12]. More recently, CART cells have been also approved for treating refractory/relapsed multiple myeloma (MM). These cells target the B cell maturation antigen (BCMA) which is highly expressed on malignant plasma cells. Two anti-BCMA second-generation CAR T cell products are currently FDA-approved: idecabtagene vicleucel [13], with ORRs of up to 76% [14,15], and ciltacabtagene autoleucel [16], with ORRs of up to 97% [17,18].

Along with these significant benefits, CAR T cells have also introduced novel toxicities, most notably, cytokine release syndrome (CRS) and immune-effector-cell-associated neurotoxicity syndrome (ICANS). Both toxicities are triggered by rapid T cell expansion, leading to the release of cytokines [1,19]. CRS is a systemic inflammatory response observed in 37–93% of patients [4] within the first week of infusion. It manifests as fever, hypotension, hypoxia, and multiple organ dysfunctions (e.g., pulmonary, renal, hepatic, gastrointestinal, and musculoskeletal) [3,19]. Neurotoxicity occurs in up to 50–65% of patients [4] and has a wide range of neurological symptoms, such as delirium, encephalopathy, dysphasia, tremors, ataxia, dysmetria, aphasia, confusion, alterations in wakefulness, hallucinations, and seizures [5,9,19]. Interestingly, products with a CD28 domain mainly cause neurotoxicity, while products with a 4-1BB domain are mostly associated with CRS [1,3,20]. The main treatment for CRS and secondary neurotoxicity is an IL-6 receptor monoclonal antibody, tocilizumab, while corticosteroids are also used in combination with tocilizumab or when treatment with tocilizumab is insufficient [4,21,22]. The chimeric anti-IL-6 monoclonal antibody siltuximab [3] and the IL-1 receptor antagonist anakinra [23] have been also suggested as alternatives, with very limited published data available.

Considering the severe toxicities that follow CAR T cell therapies, early diagnoses of both CRS and ICANS could facilitate better outcomes. Furthermore, only a limited number of studies have focused on the minority of patients who are resistant to steroids [24] and have attempted to understand their molecular and clinical characteristics. Therefore, we conducted this study to provide real-world data regarding the toxicities and efficiencies of the used therapeutic methods.

## 2. Materials and Methods

We enrolled consecutive adult patients referred to our center for CAR T therapy. The patients received lymphodepleting therapy before the CAR T therapy, with cyclophosphamide plus fludarabine according to each product’s protocol. Our group carefully implemented the EBMT and MD Anderson recommendations on eligibility and management, focusing on close collaboration with neurologists and intensive care physicians. Notably, tocilizumab was administered at an early stage, and this was accompanied by bedside EEG (electroencephalography) to guide management. This study was approved by the local institutional review board (Protocol 753/2019) in compliance with the STROBE Statement [25].

### 2.1. Toxicity Assessment

CRS and ICANS were prospectively assessed according to the American Society for Transplantation and Cellular Therapy Guidelines [26]. The Immune-Effector-Cell-Associated Encephalopathy (ICE) assessment tool was used daily during patients’ hospitalizations, concurrent with clinical monitoring. Moreover, prolonged EEG monitoring, with 20 electrodes arranged according to the international 10–20 system, was consecutively conducted in all patients both before the initiation of therapy and at days +2, +4, +6, and +14 following infusion. In the case of language dysfunction, an abnormal neurologic examination, or the presence of encephalopathy, additional EEG examinations were performed at the clinician’s request. Prophylactic anti-seizure medication was administered to all patients according to the hospital’s protocol. The review software used was the Natus Neuroworks software (latest version).

### 2.2. Statistical Analysis

We described the continuous variables as medians and ranges and the categorical variables as frequencies. We compared variables related to patients, diseases, and transplants using chi-square statistics for the categorical variables and the Mann–Whitney test for continuous variables. We calculated the probability of progression-free survival (PFS) and overall survival (OS) according to Kaplan–Meier estimates. We measured the follow-up times in order to analyze survival since the time of infusion. The analysis recorded the following factors: age, disease, previous therapies, CRP, LDH and ferritin levels at admission/infusion/day 14, leukapheresis and infusion characteristics, and short- and long-term toxicities.

## 3. Results

### 3.1. Patient Population

CAR T cells were infused in 31 patients (median age 45, range of 18–74); 2 could not receive the products due to clinical deterioration/death and 2 were scheduled for infusion. Tisagenlecleucel was infused in 12 patients, axicabtagene ciloleucel in 14, and brexucabtagene autoleucel in 5. The diagnoses were DLBCL (*n* = 16), PMBCL (*n* = 3), B-ALL (*n* = 7), and MCL (*n* = 5), and the median number of previous treatments was four (two to five).Two patients with lymphoma had undergone autologous transplantations, as had four patients with B-ALL. Autologous transplantations had been performed in one patient with DLBCL and one patient with MCL at 2.1 and 1.8 years, respectively, before the CAR T cell therapy. Allogeneic transplantations were performed in four (*n* = 4) patients with ALL at a median of 0.9 years before the CAR T cell therapy. The patient characteristics are summarized in Table 1.

### 3.2. Molecular and Clinical Characteristics

The median LDH at baseline was 236 (range of 132–2499), while the value for ferritin was 239 (range of 5–7928) mg/dL. C-reactive protein (CRP), interleukin-6, and ferritin were also measured at admission to the BMT Unit, at infusion, and on day 14. There were no significant differences in the molecular and clinical characteristics between the patients with MCL and those with other diagnoses, except for the ages being higher in the MCL patients (*p* = 0.014).

Cytokine release syndrome (CRS) was noted in 26 (grade I: 14, grade II: 6, grade III: 4, and grade IV: 2) patients, while neurotoxicity (ICANS) was noted only in 7 (grade I: 6 and grade III: 1). Tocilizumab was administered for CRS in 18 patients, while short-term, low-dose steroids were administered to the single patient who developed grade III CRS and, subsequently, grade I ICANS. High-dose steroids, along with anakinra and siltuximab, were administered to only the two MCL patients who needed to be transferred to the ICU.

### 3.3. Description of Grade III ICANS

A 63-year-old patient with relapsed/refractory MCL received conditioning with fludarabine/cyclophosphamide, followed by 2.5 × 10^8^ CAR T cells/kg. On day one, the patient presented with a fever (cytokine release syndrome (CRS) grade I). After a full diagnostic work-up, empirical treatment with antibiotics was started. On days two and three, he was asymptomatic. On day four, he was again febrile, and then pancytopenic. The antibiotics were changed according to the isolated *K. pneumoniae*. Due to his persistent fever, tocilizumab treatment commenced. Supportive treatment was provided for tumor lysis syndrome, which was diagnosed and treated according to standard practice in accordance with the relevant international standards [27]. On day six, the patient gradually developed breathing difficulties and hypoxia (SpO2:92% and FiO2: 21%). He was supported with oxygen in the nasal cannula (CRS grade II). Immune-effector-cell-associated neurotoxicity (ICANS) grade I manifested as lethargy and writing disorders (ICE: 9/10). Oral levetiracetam, which had been administered prophylactically since the day of infusion, was replaced with an increased dose of intravenous levetiracetam, along with dexamethasone, due to bedside EEG alterations. On day seven, the patient was hypotensive and did not respond to intravenous fluids (CRS grade III). Norepinephrine treatment commenced and he was intubated in the ICU. His neutrophils recovered, along with very high levels of interleukin-6 and measurable circulating CAR T cells (Figure 1). High-dose intravenous methylprednisolone and anakinra were administered. Bedside EEG and magnetic resonance imaging (MRI) confirmed grade III ICANS. Siltuximab was also administered, along with a supportive treatment. After 34 days, he was extubated without neurologic deficits but with myopathy and the colonization of resistant Gram-negative bacteria. One month after the CAR T cell therapy, complete remission (CR) was achieved, and the patient was discharged.

### 3.4. Outcomes

Despite the presence of B cell aplasia in all patients, no infections requiring hospitalizations were noted. This was particularly important because of the increased safety measures (negative PCR testing) required for hospitalization since the beginning of the COVID-19 pandemic. With a median follow up of 13.4 (1.4–29.1) months, nine patients were then in CR. The progression-free (PFS) and overall survival (OS) rates were 41.2% and 88.1%, respectively, at one year. As expected, age was not associated with PFS or OS. Among the baseline characteristics, MCL diagnosis, which coincides with the administration of brexucabtagene autoleucel, was the only factor associated with a poor OS (a one-year OS of 93.3% versus 0%, *p* < 0.001, Figure 1); meanwhile, an increased LDH was associated with PFS (*p* = 0.027). In addition, CRP at day 14 was associated with a poor OS (*p* = 0.001).

## 4. Discussion 

In conclusion, our study confirmed that CAR T cell therapy has revolutionized the field of hematologic malignances and has increased the overall survival rate, even leading to full remission where other treatments have failed. As indications and products expand, a combination of molecular tests, clinical evaluations, and regular bedside evaluations is needed to diagnose toxicities and treat them effectively.

Long-term real-world experience has been gained regarding the two first products used in novel CAR T cell therapies: tisagenlecleucel and axicabtagene ciloleucel. Tisagenlecleucel is a 4-1BB co-stimulatory domain-based CAR T that first received FDA approval for use in the treatment of relapsed/refractory pediatric and young adult patients with B-cell lineage ALL after the results of a phase 2 study (ELIANA) [28]. Later, it was also approved for treating patients with relapsed/refractory DLBCL after the results of another phase 2 study (JULIET) [5,29]. On the other hand, axicabtagene ciloleucel is a CD28 domain-based CAR T that received FDA approval after the results of the pivotal ZUMA-1 study on the treatment of refractory large B-cell lymphoma [9,30]. More specifically, the ELIANA study for ALL showed an overall response rate (ORR) of 66% while the updated analysis [31] showed an ORR of 81%. In the same study, 77% of the patients experienced CRS (46% grades III and IV) and 40% experienced neurotoxicity (13% grade III) [28]. In the PMR study conducted by the Center for International Blood and Marrow Transplant Research (CIBMTR), the complete remission rate was 85.5%. CRS was experienced by 55% of patients (16.1% grade ≥ III) and ICANS was experienced by 27% of patients (9% grade ≥ III) [26,32,33]. For the treatment of large B-cell lymphoma, the JULIET study showed an ORR of 52% with the use of tisagenlecleucel while grade ≥III ICANS was observed in 12% of cases [29]. The PMR study showed that the ORR of patients treated with tisagenlecleucel was 61.8% while CRS occurred in 5% of patients (4.5% grade ≥ III) and ICANS in 18% of patients (5.5% grade ≥ III) [32]. Lastly, the ZUMA-1 study showed that the use of axicabtagene ciloleucelled to an ORR of 82% while grade ≥III ICANS was observed in 31% of patients [9].By examining these results more closely, we could see that, in the treatment of large B-cell lymphoma, axicabtagene ciloleucel is associated with both higher efficacy and higher toxicity than tisagenlecleucel.

CRS and ICANS are the two main adverse reactions related to CAR T cell therapies, and they both present with a variety of severe and potentially dangerous symptoms. Both toxicities are associated with the rapid activation of CAR T cells, the CAR T cell dose, and the pretreatment BM tumor burden [1,21]. More specifically, CAR T cell expansion causes T cells to secrete cytokines, which leads to the activation of other cell types, such as monocytes and endothelial cells. These immune cells release more pro-inflammatory cytokines, creating a cytokine storm, and this seems to be the main cause of CRS [34,35,36,37]. Patients with severe CRS appear to have higher serum concentrations of IL-6, other cytokines (e.g., INF-γ, IL-5, IL-10, and GM-CSF), and C-reactive protein (CRP; ≥20 mg/dL) [38]. On the other hand, the exact pathogenesis of neurotoxicity is still unknown. CRS is the main risk factor for ICANS since most cases occur after the complete resolution of CRS, suggesting a potential mechanistic link [22,39]. This link has been further confirmed by the fact that the clinical correlates of severe ICANS often overlap with severe CRS, including elevations in C-reactive protein (CRP), ferritin, and cytopenias. High levels of cytokines such as IL-6 and CXCL10 in the cerebrospinal fluid (CSF) can also be associated with ICANS, but it is unclear whether they are caused by a breakdown in the blood–brain barrier (BBB) or are directly released from CNS resident cells after infusion [22,40,41,42]. Finally, another possible cause is the direct toxicity from the CAR T cells that target brain cells that express CD19 [43,44].

A newly recognized entity that must be distinguished from CRS is immune-effector-cell-associated hemophagocytic lymphohistiocytosis (HLH)-like syndrome (IEC-HS). IEC-HS is a hyperinflammatory condition caused by the activation of macrophages from tumor-activated CAR T cells, and it leads to a cytokine release, creating a positive feedback loop [45]. IEC-HS is mostly associated with coagulopathy, cytopenias, hyperferritinemia, hypertriglyceridemia, fever, hepatosplenomegaly, neurologic toxicities, and organ failure [46,47,48]. Although similar HLH-like symptoms are present in patients who experience severe CRS [26,49,50,51], IEC-HS is often delayed and manifests later, after the CRS is resolved [50,52]. Interestingly, an elevation in ferritin levels is required for this diagnosis since normal ferritin levels have a negative predictive value [48,53]. Bone marrow biopsies in patients with HLH have shown that both hemophagocytic activity and cellularity are increased, the bone marrow appears foamier, and the histiocyte infiltrates are elevated. Treatments should be selected carefully, with the first-line agents being anakinra [50] and corticosteroids [54,55] while the second-line treatments should include ruxolitinib [56], etoposide [57], and emapalumab [50]. Regardless of their differences, both CRS and IEC-HS are potentially life-threatening conditions; for this reason, more research needs to be conducted in order to fully elucidate the mechanisms leading to these toxicities. It is undeniable that the more we learn, the better prepared we will be to distinguish, predict, and treat these conditions.

## 5. Conclusions

To our knowledge, our results are representative of current clinical practices in many centers, contributing data on the optimal bedside diagnosis of neurological manifestations, the early use of tocilizumab, and the treatment of refractory cases.

Our study is limited by its retrospective nature, the relatively small number of participants, and the fact that it reflects the experience of only one center. Nevertheless, it reflects the local epidemiology and reports data from an adult hematology center located in Northern Greece. However, it should be noted that we conducted this study according to the standard operating procedures with long-term follow-up examinations despite difficulties related to COVID-19. Further studies are needed to determine personalized approaches depending on products and indications.

## Figures and Tables

**Figure 1 cancers-15-04253-f001:**
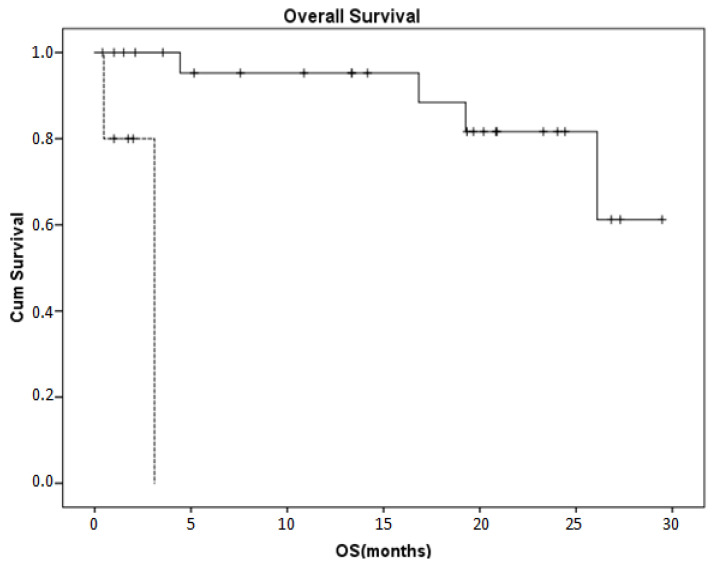
Kaplan–Meier curve of the overall survival rate in our patient population. An MCL diagnosis that coincided with the administration of brexucabtagene autoleucelinin five patients (dotted line)was the only factor associated with a poor OS compared to the other diagnoses/products in 26 patients (*p* < 0.001).

**Table 1 cancers-15-04253-t001:** Molecular and clinical characteristics of the patients.

Diagnosis	Number of Patients
DLBCL	16
PMBCL	3
B-ALL	7
MCL	5
Patient characteristics	Results
Age	45 (range of 18–74)
LDH (IU/L)	236 (range of 132–2499)
Ferritin (mg/dL)	239 (range of 5–7928)
Previous autologous transplantation	2
Previous allogeneic transplantation	4
Treatment	Number of patients
Tisagenlecleucel	12
Axicabtagene ciloleucel	14
Brexucabtagene autoleucel	5

DLBCL, diffuse large B-cell lymphoma; PMBCL, primary mediastinal large B-cell lymphoma; B-ALL, B-cell acute lymphoblastic leukemia; MCL, mantle cell lymphoma; LDH, lactate dehydrogenase.

## Data Availability

The data will immediately be made available upon request to the corresponding author.

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
