# Peer review of "Molecular and Clinical Characteristics of Different Toxicity Rates in Anti-CD19 Chimeric Antigen Receptor T Cells: Real-World Experience"

_cancers, 2023, doi:10.3390/cancers15174253_

Round 1

Reviewer 1 Report

If possible, cite references according to the journal schema, in order to obtain a consequent abbreviation.  It would be interesting to deepen the macrophages activation syndrome, eventually with direct morphological data (bone marrow biopsy, peripheral blood smear, etc), or concerning their lineage monocytes correlated. 

Author Response

Point 1: If possible, cite references according to the journal schema, in order to obtain a consequent abbreviation. 

Response 1: We thank the reviewer for the suggestion. We cited the references according to the journal scheme.

Point 2: It would be interesting to deepen the macrophages activation syndrome, eventually with direct morphological data (bone marrow biopsy, peripheral blood smear, etc), or concerning their lineage monocytes correlated. 

Response 2: We appreciate the reviewer's comment and have added additional data in the text.

Reviewer 2 Report

This is an original research article about the real-life experience of 31 adult patients with hematological malignancies (DLBCL, PMBCL, B-ALL, MCL) treated with anti-CD19 chimeric antigen receptor T (CAR-T) cells (Tisagenlecleucel, Axicabtageneciloleucel, Brexucabtagene autoleucel) and their toxicity rates, i.e. cytokine release syndrome (CRS) and immune effector cell-associated neurotoxicity syndrome (ICANS).  Of 31 patients, 26 developed CRS (6 had grade 3-4) and 7 developed ICANS, and were treated with immunosuppressive therapy (Tocilizumab or Siltuximab (anti-IL-6 antibodies) or Anakinra (IL-1R antagonist)).  After a median follow-up of 13 months, 9 patients are in complete remission (CR). Progression-free survival (PFS) and overall survival (OS) were 41% and 88%, respectively.  Treated MCL patients had poor OS.

Although this article presents valuable clinical data, the manuscript is incomplete and difficult to understand at times.  The laboratory methods used and the assessment of CRS (grade 1-4) are not described in the materials and methods section. In the results section, the patient populations, clinical and molecular characteristics, toxicity profiles and patient outcomes are not described in adequate detail, including numerical and statistical data.  The results are not adequately discussed in the context of previous works.

Line 21-50: the text of the abstract is much too long, and should be more concise and focused.

Line 24: what were the indications (DLBCL, PMBCL, B-ALL, MCL) for CAR-T cell therapy?  Did the non-MCL patients who experienced toxicity respond to immunosuppressive therapy?  Indicate the magnitude of effects of PFS and OS, not just the p-values.

Line 51-52: add cytokine release syndrome (CRS) and immune effector cell-associated neurotoxicity syndrome (ICANS) to the keywords

Line 65-77: which CAR-T cell therapies are first, second or third generation? 

Line 81-89: describe in more detail the cellular and molecular mechanisms of CRS and ICANS, including the names of cytokines involved.

Line 104-135: how was the CRS assessed?  Other than LDH and ferritin, were other criteria assessed (e.g., prolonged fever, organ dysfunction, blood count, electrolytes, cytokine profiles, immune function markers, etc.)?  Were conditions such as diabetes, hypertension or obesity associated with CRS in your study?

Line 136-159, 182-190: describe in more detail the patient populations, clinical and molecular characteristics, toxicity profiles and patient outcomes. Include additional figures or tables as needed.  For which patients did the COVID-19 pandemic interfere with  follow-up?

Line 167: how was tumor lysis syndrome diagnosed and treated?

Line 195-267: rather than presenting a review of the pertinent literature, discuss how your patient results compare or differ with previous reports.  Highlight the aspects of your work that are original contribution to knowledge.  Shorten the section about immune effector cell-associated hemophagocytic lymphohistiocytosis-like syndrome as it is not discussed elsewhere in your text.

Table 1: what is the pathological stage of each patient?  Which patients underwent autologous or allogeneic transplantation, and how long before CAR-T cell therapy?

Fig. 1: clearly label which patients and how many patients correspond to the solid line (? DLBCL, PMBCL, B-ALL) or the dotted line (MCL diagnosis).

Overall, the manuscript contains numerous spelling and grammatical errors, and incorrect word usage and sentence structure.  Key information is often not provided in the text where needed, which renders the text difficult to understand.  Define all abbreviations/acronyms at their first utilization in the abstract and the main text.  Extensive revision is necessary.

Author Response

Point 1: Line 21-50: the text of the abstract is much too long, and should be more concise and focused.

Response 1:We thank the reviewer for the comment. We have fixed the length of the abstract.

Point 2:Line 24: what were the indications (DLBCL, PMBCL, B-ALL, MCL) for CAR-T cell therapy?  Did the non-MCL patients who experienced toxicity respond to immunosuppressive therapy?  Indicate the magnitude of effects of PFS and OS, not just the p-values.

Response 2:Indications are shown in Table 1. We have added them in the revised text. Non-MCL patients responded well to therapy and therefore, had better overall survival. We have also added and OS values in the comparison between groups. The other associations were shown with continuous variables, and therefore exact values cannot be shown.

Point 3: Line 51-52: add cytokine release syndrome (CRS) and immune effector cell-associated neurotoxicity syndrome (ICANS) to the keywords

Response 3: We thank the reviewer for the recommendation. We added the keywords

Point 4: Line 65-77: which CAR-T cell therapies are first, second or third generation? 

Pesponse 4: We inform the reviewer that all generations are explained in the first paragraph of the introduction. We added the generation of every CAR-T cell therapy we mention.

Point 5: Line 81-89: describe in more detail the cellular and molecular mechanisms of CRS and ICANS, including the names of cytokines involved.

Response 5: We thank the reviewer for the suggestion. The exact cellular and molecular mechanisms as well as the cytokines involved are described in the discussion.

Point 6: Line 104-135: how was the CRS assessed?  Other than LDH and ferritin, were other criteria assessed (e.g., prolonged fever, organ dysfunction, blood count, electrolytes, cytokine profiles, immune function markers, etc.)?  Were conditions such as diabetes, hypertension or obesity associated with CRS in your study?

Response 6: We thank the reviewer for giving us the opportunity to clarify this issue. CRS and ICANS were assessed according to the consensus criteria. Other comorbidities were not associated with CRS in our study.

Point 7: Line 136-159, 182-190: describe in more detail the patient populations, clinical and molecular characteristics, toxicity profiles and patient outcomes. Include additional figures or tables as needed.  For which patients did the COVID-19 pandemic interfere with  follow-up?

Response 7: We have clarified in the revised manuscript our comment on COVID-19. Our center operates with increased safety measures requiring PCR testing for all hospitalized patients during the whole period up to now.

Point 8: Line 167: how was tumor lysis syndrome diagnosed and treated?

Response 8: Diagnosis and treatment was performed according to standard practice in accordance to international standards. Our center is JACIE-accredited since 2013 and all practices are carefully reviewed.

Point 9: Line 195-267: rather than presenting a review of the pertinent literature, discuss how your patient results compare or differ with previous reports.  Highlight the aspects of your work that are original contribution to knowledge.  Shorten the section about immune effector cell-associated hemophagocytic lymphohistiocytosis-like syndrome as it is not discussed elsewhere in your text.

Response 9: We thank the reviewer for giving us the opportunity to improve our manuscript. To our knowledge, our results are representative of current clinical practice in many centers, adding data on optimal bedside diagnosis of neurological manifestations, early use of tocilizumab, and treatment of refractory cases. We have shorten the paragraph about IEC-HS .

Point 10: Table 1: what is the pathological stage of each patient?  Which patients underwent autologous or allogeneic transplantation, and how long before CAR-T cell therapy?

Response 10: We are not sure what is meant by “pathological stage”. All patients suffered from relapsed/refractory disease with pathology defining entities according to indications. We have added details of transplant in the text.

Point 11: Fig. 1: clearly label which patients and how many patients correspond to the solid line (? DLBCL, PMBCL, B-ALL) or the dotted line (MCL diagnosis).

Response 11: The number of patients is shown in Table 1. Prompted by the reviewer’s comment, we have also added it in the legend.

Point 12: Overall, the manuscript contains numerous spelling and grammatical errors, and incorrect word usage and sentence structure.  Key information is often not provided in the text where needed, which renders the text difficult to understand.  Define all abbreviations/acronyms at their first utilization in the abstract and the main text.  Extensive revision is necessary.

Response 12: We have tried to extensively revise our manuscript and have also performed MDPI language editing.

Reviewer 3 Report

The article entitled "Molecular and Clinical Characteristics of Different Toxicity Rates in anti-CD19 Chimeric Antigen Receptor T Cells: Real-World Experience" submitted by E. Gavriilaki et al. is interesting, novel and much needed study to understand and treat the cytokine release syndrome (CRS) and neurotoxicity (ICANS) in CAR T cell therapy. I would like to recommend to publish the work in the journal Cancers in present form.

Best

Author Response

Point 1: The article entitled "Molecular and Clinical Characteristics of Different Toxicity Rates in anti-CD19 Chimeric Antigen Receptor T Cells: Real-World Experience" submitted by E. Gavriilaki et al. is interesting, novel and much needed study to understand and treat the cytokine release syndrome (CRS) and neurotoxicity (ICANS) in CAR T cell therapy. I would like to recommend to publish the work in the journal Cancers in present form.

Response 1: We thank the reviewer for helping us improve our manuscript.

Round 2

Reviewer 2 Report

Line 21-51.  According to the Instructions for Authors of /Cancers/, "in
all manuscript types… the abstract should be a total of about 200 words
maximum". The current word count of your Abstract is 366 words. This is
in addition to the "Simple Summary", which is required for the journal
Cancers. Your reply does not fulfill the Journal requirements for the
Abstract. Please rewrite and shorten the Abstract.

Line 105-106.  Cite in the text the “limited number of studies…” mentioned.

Words are sometimes merged together. For example, "transplantationswereperformed' (line152).

Author Response

Q1: Line 21-51.  According to the Instructions for Authors of /Cancers/, "in
all manuscript types… the abstract should be a total of about 200 words
maximum". The current word count of your Abstract is 366 words. This is
in addition to the "Simple Summary", which is required for the journal
Cancers. Your reply does not fulfill the Journal requirements for the
Abstract. Please rewrite and shorten the Abstract.

R1: We thank the reviewer for the comment. We have limited the number of words in our Abstract

Q2: Line 105-106.  Cite in the text the “limited number of studies…” mentioned.\

R2: We have added the requested citations

Q3: Words are sometimes merged together. For example, "transplantationswereperformed' (line152).

R3: We have tried to correct the merged words. We have also performed MDPI language editing.